# Modeling Chikungunya control strategies and Mayaro potential outbreak in the city of Rio de Janeiro

Esteban Dodero-Rojas[1,2], Luiza G. Ferreira[3], Vitor B. P. Leite[4], José N. Onuchic[1,3,5,6], Vinícius G. Contessoto[1,7]*

**1** Center for Theoretical Biological Physics, Rice University, Houston, TX, United States of America, **2** Theoretical and Computational Physics Laboratory, University of Costa Rica, San José, Costa Rica, **3** Department of Chemistry, Rice University, Houston, TX, United States of America, **4** Department of Physics, Institute of Biosciences, Letters and Exact Sciences, São Paulo State University - UNESP, São José do Rio Preto, SP, Brazil, **5** Department of Physics & Astronomy, Rice University, Houston, TX, United States of America, **6** Department of Biosciences, Rice University, Houston, TX, United States of America, **7** Brazilian Biorenewables National Laboratory - LNBR, Brazilian Center for Research in Energy and Materials - CNPEM, Campinas, SP, Brazil

\* contessoto@rice.edu

**Data Availability Statement:** All relevant data are within the paper and its Supporting Information files.

## Abstract

Mosquito-borne diseases have become a significant health issue in many regions around the world. For tropical countries, diseases such as Dengue, Zika, and Chikungunya, became epidemic in the last decades. Health surveillance reports during this period were crucial in providing scientific-based information to guide decision making and resources allocation to control outbreaks. In this work, we perform data analysis of the last Chikungunya epidemics in the city of Rio de Janeiro by applying a compartmental mathematical model. Sensitivity analyses were performed in order to describe the contribution of each parameter to the outbreak incidence. We estimate the "basic reproduction number" for those outbreaks and predict the potential epidemic outbreak of the Mayaro virus. We also simulated several scenarios with different public interventions to decrease the number of infected people. Such scenarios should provide insights about possible strategies to control future outbreaks.

## Introduction

In the last decades, Mosquito-borne diseases have become a significant health issue in many regions around the world. Projections indicate that around 2050, half of the population will be at risk of some arbovirus infection [1]. These arboviruses, which include diseases such as Dengue, Zika, and Chikungunya, are epidemic in most of the tropical countries. Besides temperature and humidity, human migrations and sanitation also contribute to the epidemic conditions in these places [2, 3]. For example, around 300.000 people were infected by Dengue, Zika, or Chikungunya by the end of the 11[th] week of 2019 in Brazil. This number represents almost three times the reported cases in 2018 for the same period [4]. These surveillance reports over time are essential in providing scientific-based information to guide decision making, resources allocation, and interventions [5]. The usage of mathematical models has demonstrated to be a powerful tool in contributing to these data analysis [6–8]. One of the most significant parameters

**Funding:** This work was supported by the Center for Theoretical Biological Physics sponsored by the NSF (https://www.nsf.gov) (Grant PHY- 1427654) and by NSF-CHE 1614101. JNO is a Cancer Prevention and Research Institute of Texas (CPRIT) Scholar in Cancer Research. VGC was funded by Grant 2016/13998-8 and 2017/09662-7, FAPESP (S\~ao Paulo Research Foundation and Higher Education Personnel - http://www.fapesp.br) and CAPES and (Higher Education Personnel Improvement Coordination - http://www.capes.gov.br/). VBPL also acknowledges FAPESP Grant 2018/18668-1. The funders had no role in study design, data collection and analysis, decision to publish, or preparation of the manuscript.

extracted from these analyses is the basic reproduction number $R_o$. $R_o$ is defined as the number of secondary infections derived from one single infectious subject and is widely used as an epidemiological metric employed to describe the transmissibility of infectious agents [9].

Here we apply a compartmental mathematical model to investigate the dynamics of Chikungunya outbreaks in the city of Rio de Janeiro in Brazil. The model consists of ordinary differential equations that describe the transmission and the transition of the diseases in humans and vectors [10, 11]. The model's parameters were extracted from the literature or obtained from the best fit from the data of Rio de Janeiro surveillance report for the years of 2016, 2018 and 2019 [12]. Based on these parameters, we estimate the basic reproduction number $R_o$ for Chikungunya outbreaks in those years. We also simulate a scenario predicting if the Mayaro virus could be a potential epidemic disease in Rio de Janeiro. Modifications in the standard model equations were implemented to introduce different possible interventions in order to decrease the number of infected people [13]. Those simulated interventions include actions such as killing adult mosquitoes by fogging, decreasing mosquitoes birth rate by removing places where the vector lays eggs, e.g., removing standing water and, decreasing the contact

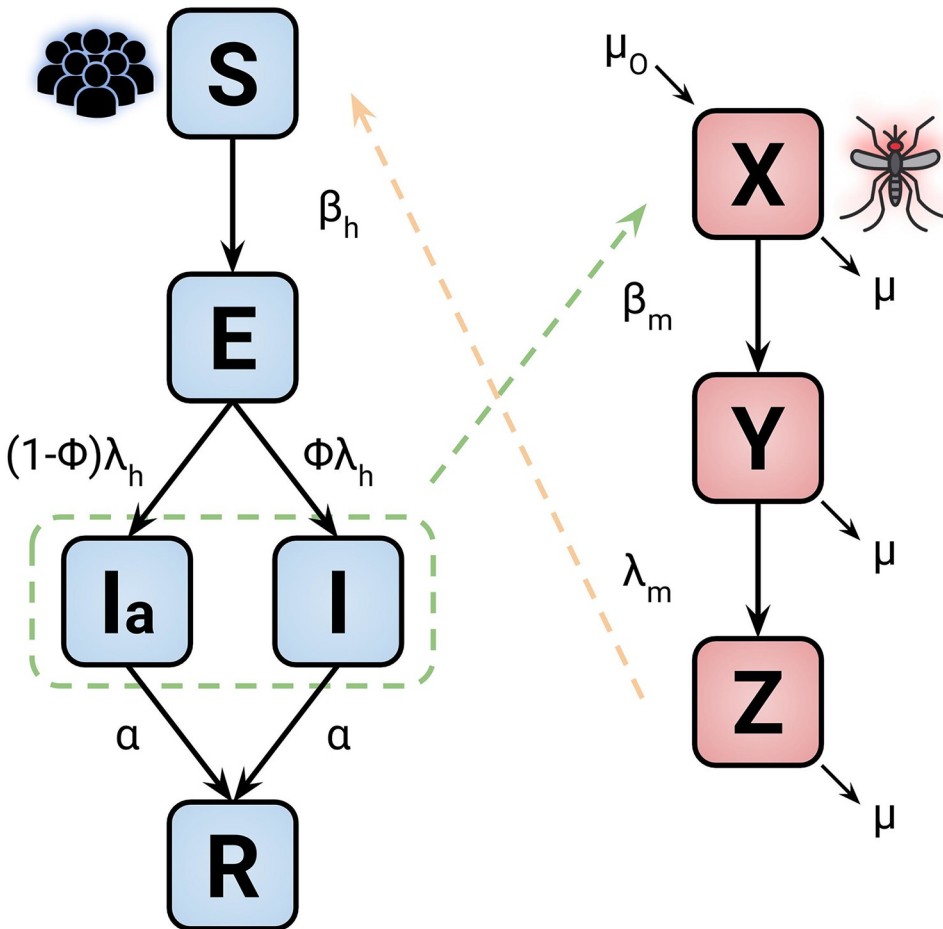

**Fig 1. Model diagram representation of the disease dynamics.** The blocks blue and red are related to human and mosquitoes, respectively. The parameter and variable descriptions are presented in Tables 1 and 2. The dashed lines represent the transmission of the disease between the two groups.

between an infected human with mosquitoes by stimulating repellent usage. A scenario containing all interventions was also performed for different intensities of those actions.

## Materials and methods

In this work, we perform mathematical modeling of Chikungunya outbreaks in Rio de Janeiro for the years 2016, 2017 and 2019 [12]. The Chikungunya virus infects humans through mosquitoes as the disease vector. The model adopted here is a compartmental model known as SEIR (Susceptible, Exposed, Infected, and Recovered) [8, 10, 14]. The approaches using this class of models have been successful in modeling epidemic related to human vector dynamics [11, 15]. Fig 1 presents a schematic description of this modeling.

The human disease flow is presented by the blue blocks where $S$ is the susceptible proportion of humans, which become exposed to the virus, $E$, at a rate $\beta_h$ after the contact with infectious mosquitoes $Z$. After the latent period $\lambda_h$, the exposed humans become infectious: either symptomatically $I$ or asymptomatically $I_a$; the parameter $\phi$ determines the ratio between the infectious states. Finally, the infected humans recover reaching the state $R$ at rate $\alpha$.

In the case of the vectors disease flow, shown by the red blocks in Fig 1, the susceptible mosquitoes $X$ become exposed $Y$ at a rate $\beta_m$ after acquiring the virus from infectious humans. $\lambda_m$ defines the latent period for the exposed mosquito to transition to the infectious state $Z$. In this modeling, we assume that human mortality and birth rates are the same, keeping the human population constant. For the vectors, we set the parameter $\mu$ and $\mu_o$ as the mortality and birth rate, respectively. The model is represented by the following set of differential equations:

$$\frac{dS}{dt} = -\beta_h SZ$$

$$\frac{dE}{dt} = \beta_h SZ - \lambda_h E$$

$$\frac{dI}{dt} = \phi \lambda_h E - \alpha I$$

$$\frac{dI_a}{dt} = (1 - \phi)\lambda_h E - \alpha Ia$$

$$\frac{dR}{dt} = \alpha(I + I_a)$$

$$\frac{dX}{dt} = \mu_o - \beta_m X(I + I_a) - \mu X$$

$$\frac{dY}{dt} = \beta_m X(I + I_a) - \lambda_m Y - \mu Y$$

$$\frac{dZ}{dt} = \lambda_m Y - \mu Z$$

Table 1 shows the definition of each state in the model for both humans and mosquitoes. Those states will dynamically vary during the model simulation in which the parameter $I$ related to the number of cases reported by the surveillance data will be the variable used in the fitting of the model simulation.

**Table 1. Definition of the state variables used in the model for both humans and mosquitoes.**

| Symbol | Definition |
|--------|------------|
| $S$ | Susceptible proportion of human population |
| $E$ | Exposed proportion of human population |
| $I$ | Symptomatically infectious proportion of human population |
| $I_a$ | Asymptomatically infectious proportion of human population |
| $R$ | Recovered proportion of human population |
| $X$ | Susceptible proportion of mosquito population |
| $Y$ | Exposed proportion of mosquito population |
| $Z$ | Infectious proportion of mosquito population |

Table 2 describes the parameters and ranges used in the model. Some data information comes from the literature, and for the parameter which we have no description, they will be obtained from the model best fit.

In this work we estimate the basic reproduction number $R_o$ by applying the next generation matrix method [21, 22]. $R_o$ indicates the number of secondary infections derived from one single infectious subject and can be described by:

$$R_o = -\rho(\Pi\Gamma^{-1}) \tag{1}$$

Where $\rho(K)$ is the spectral radius of the matrix $K = \Pi\Gamma^{-1}$. $\Pi$ is the transmission matrix that contains the rates of humans to get infected by the vector and vice-versa:

$$\Pi = \begin{pmatrix} 0 & 0 & 0 & \beta_h \\ 0 & 0 & 0 & 0 \\ 0 & \beta_m & 0 & 0 \\ 0 & 0 & 0 & 0 \end{pmatrix} \tag{2}$$

**Table 2. Description of the parameters and range used in the model simulation.**

| | Definition | Range (days) |
|---|------------|--------------|
| $\beta_h$ | Proportional rate at which humans get infected | Unknown |
| $\beta_m$ | Proportional rate at which mosquitoes get infected | Unknown |
| $1/\lambda_h$ | Human latent period of infection | 2-10 [16, 17] |
| $1/\lambda_m$ | Mosquito latent period of infection | 2-6 [18, 19] |
| $\alpha$ | Rate of recovery | 1-7 |
| $\mu_o$ | Mosquito birth rate | 0.05-0.03 |
| $\mu$ | Mosquito mortality rate | 0.05-0.03 |
| $\phi$ | Asymptomatically-Symptomatically infectious ratio | 0.72-0.97 [16, 20] |

$\Gamma$ is the transition matrix that takes into account the transitions from being exposed to become infectious:

$$\Gamma = \begin{pmatrix} -\lambda_h & 0 & 0 & 0 \\ \lambda_h & \alpha & 0 & 0 \\ 0 & 0 & -(\mu + \lambda_m) & 0 \\ 0 & 0 & 0 & -\mu \end{pmatrix} \qquad (3)$$

The mathematical solution of (1) gives an expression [11]:

$$R_o = \sqrt{\frac{\beta_h \beta_m \lambda_m}{\mu \alpha (\mu + \lambda_m)}} \qquad (4)$$

The Eq 4 has parameters in which there is no information available such as $\beta_h$ and $\beta_m$. In order to estimate $R_o$, these parameters will be obtained from the best fit of the model simulations using data from the surveillance reports [12].

## Results and discussion

The usage of the SEIR model to investigate diseases epidemics provides a tool to quantify different parameters in outbreaks. The basic reproduction number $R_o$ is the most important quantity, and it is defined as the number of secondary infections caused by an infected individual [3, 23]. It estimates the potential of an outbreak to occur in the case of $R_o > 1$ [10, 24]. The knowledge of $R_o$ also gives insights into the understanding of the epidemiology of a particular disease and its spreading changes over time and geography [25].

The number of secondary infections in humans from an infected human, defined as $R_T$, the *type reproduction number* [26], can be obtained by $R_o$ squared ($R_T = R_o^2$) [11]. The information of $R_T$ can be used to estimate the number of people that need to be isolated or vaccinated ($Q$) to contain the epidemic using the relation $Q = 1 - 1/R_T$. We present the application of the SEIR model in the data of Chikungunya in Rio de Janeiro—Brazil in different years (2016, 2018 and 2019). We also provide an estimation of the potential outbreak of Mayaro virus in Rio de Janeiro. Some additions on the model are also proposed in a way to simulated possible interventions in the epidemic control [13, 27].

### Basic reproduction number—$R_o$

The data which contains the weekly number of infected people of Chikungunya outbreak in Rio de Janeiro was obtained from the surveillance report for the years 2016, 2018, and 2019, which are publicly available [12]. The ratio between the number of reported cases and the total population is presented in red circles Fig 2. The total number of cases reported in 2019 is 22896 until the 26th week when the data was collected. This number is almost two times higher in comparison with 2016 and 2018 in 52 weeks, 14203 and 10700, respectively. In 2017 the total number of cases was 1870, which will not be used in this study. Then, the SEIR model was applied to fit the incidence data where the upper and the lower bound of the model parameters were set to vary in a range described in the literature. The transmissions coefficients $\lambda_h$, $\lambda_m$, $\alpha$, $\beta_h$ and $\beta_m$ values are obtained from the model best-fit since there are no values reports in literature about these parameters for Rio de Janeiro [11]. The simulated number of cases from the best-fit are presented as bars in Fig 2A, 2B and 2C for the years 2016, 2018, and 2019, respectively. The least square error obtained for 2016 was of 0.0186, while for 2018 and 2019

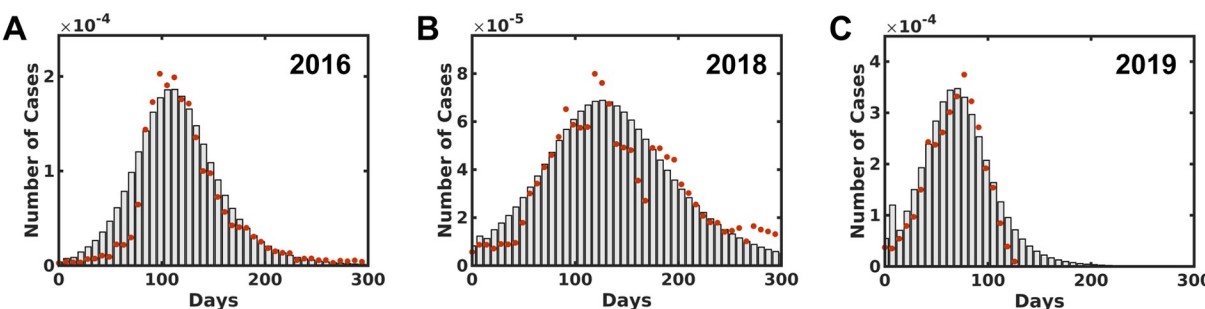

**Fig 2.** The number of infected people distribution by Chikungunya in Rio de Janeiro for 2016, 2018 and 2019, A, B, and C, respectively. The red dots describe the informed data obtained from the surveillance report [12] per week. The bars indicate the simulation best fit distribution compared to the reported data.

this measure reached 0.032 and 0.331 respectively. The parameters obtained from the fitting and used to plot Fig 2 are described in Table 3. Table 3 also shows in the last row, the estimation of $R_o$ using Eq (4) for each investigated year.

All the estimated $R_o$ values for the three studied years are greater than one describing how severe were the outbreaks in which 2019 has the highest value $R_o^{2019} = 1.95$ compared with $R_o^{2016} = 1.82$ and $R_o^{2018} = 1.38$. Considering other epidemic diseases in Rio de Janeiro as Dengue and Zika, which are transmitted by the same vector, the parameters estimated here are similar to other registered outbreaks studies [25, 28]. It is worthwhile to mention that these parameters give insights from a city as a whole, which is invariant on how heterogeneous the sanitary conditions could be at different neighborhoods [7, 25, 29, 30].

The last column in Table 3 presents the estimated parameters for Mayaro virus using the data for the Chikungunya outbreak in 2018, which is the most recent complete data available. The assumption is based on the similarities between these two *alphavirus* in which they can also be transmitted by the same mosquito vector: *Aedes aegypti* [6, 31, 34–36]. The estimated $R_o^{MAYV}$ for Mayaro presents values between 1.18 and 3.51 for the lower and upper limits. Even the lower bound $R_o^{MAYV}$ is greater than 1, suggesting that Mayaro has the potential to be an epidemic disease as recent reports are signaling for different locations [37–39].

## Sensitivity analysis

The usage of the SEIR model allows us to estimate the importance of each parameter in the characterization of an outbreak by performing a parameter sensitivity analysis (Fig 3). It was carried out a hundred thousand Monte Carlo simulations sampling around the ±5% range from the best fit value of the Chikungunya epidemic data of 2018. From these simulations, the time-shift on the peak and the total number of infected people were obtained. Additionally,

**Table 3. Best fitted parameter values in different years for Chikungunya (CHKV) in Rio de Janeiro.** Estimated parameter are also presented in the last column for Mayaro virus. The last row shows the values of the estimated $R_o$ for both Chikungunya and Mayaro.

| Parameter | CHKV—2016 | CHKV—2018 | CHKV—2019 | Mayaro |
|---|---|---|---|---|
| $\beta_h$ | 0.1 | 0.1 | 0.194 | 0.1–0.194 |
| $\beta_m$ | 0.732 | 0.562 | 0.298 | 0.298–0.732 |
| $\lambda_h$ | 0.17 | 0.5 | 0.17 | – |
| $\lambda_m$ | 0.17 | 0.181 | 0.17 | 0.17–0.33 [31, 32] |
| $\alpha$ | 0.343 | 0.464 | 0.235 | 0.2–0.33 [33] |
| $R_o$ | **1.82** | **1.38** | **1.95** | **1.18–3.51** |

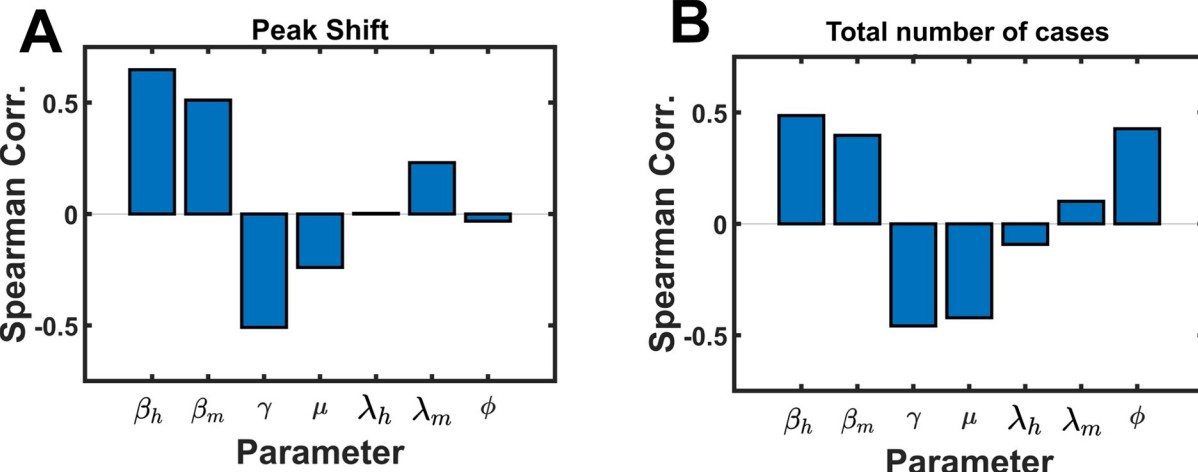

**Fig 3. Monte Carlo simulations sensitivity analysis.** A present the Spearman correlation of each parameter value with the peak shift on time and B shows the Spearman correlation of each parameter value with the total number of infected people. The values were obtain by 100.000 Monte Carlo simulations around ±5% the best fit values the 2018 outbreak.

the Spearman correlation for each parameter with the time-shift and the total number of infected people was calculated. [11]. In Fig 3 the transmission parameters ($\beta_h$ and $\beta_m$) in both: peak shift and total population infected, exhibit as an important contribution to the outbreak characterization [40]. Fig 3A shows that the rate of recovery plays a significant role in the time scale of the epidemic. The parameter presented a negative correlation, which means that positive variations in this parameter will modify the shift by delaying the peak reaching point; the same is observed about the mosquito birth-death rate. On the other hand, Fig 3B shows that by decreasing the magnitude of the transmission coefficients or increasing the value of $\gamma$ and $\mu$ may reduce the number to total cases of infected people. These parameters modification can be related by triggering public interventions, which will be discussed in the next section. Intuitively, a fast recovery would decrease the probability of a mosquito to bit an infected person, while the increase on the birth-death rate of the mosquito will create similar effect: an infected mosquito would die quickly, therefore it may not be able to transmit the disease. Sensitivity analyses based on the outbreak time dependence are presented in the Supporting Information.

## Interventions

In this section, we will discuss the outcome of different possible intervention strategies to control the epidemic disease spreading [13, 23]. The simulations were carried out using the Chikungunya epidemic outbreak data from Rio de Janeiro in 2018. The first approach simulates the action of killing adult mosquitoes, which is related to the use of insecticide as fogging. In the model, this strategy appears as an increase in the mosquitoes mortality rate $\mu$ presented in Eq 5:

$$\mu(C, t) = \mu_c (1 - \omega\, \theta(C - C_p)) \tag{5}$$

where $\theta(\ldots)$ is the unitary step function, $\mu_c$ is the natural rate of birth/death of the mosquito, $C$ is the cumulative number of infected people and, $\omega$ is the parameter related to the intensity of the fogging action reflected in the mosquito death rate $\mu$. The fogging action is triggered when the cumulative number of infected people $C$, described in Eq 6, reaches the value $C_p$ which is 30% of the total number of cases from the real data. For all the interventions discussed in this

study, the trigger event will be the same as the one present here in the fogging action.

$$\frac{dC}{dt} = \phi \lambda_h E \qquad (6)$$

Fig 4 presents the distribution of the number of cases and the cumulative number of cases as a function time for different fogging intensities, A and B, respectively. In Fig 4A, once the fogging action starts, the number of cases per week stop to grow and starts to decrease over time. The strength of the parameter $\omega$ dictates how fast these curves decay. The cumulative number of cases also reflects the fogging action for different intensities, as presented in Fig 4B. The total number of case drops to 70.0% when $\omega = 0.25$ presented in dashed red line and drops to 54.0% when $\omega = 0.5$ shown in the dotted yellow line when compared with the real data without fogging action $\omega = 0.0$.

The second simulated intervention is the action of reducing the birth rate of the vector. This approach can be associated with population orientation or better social sanitary conditions. These actions may produce a decrease in the number of places where the mosquitoes lay the eggs such as standing water, for example. Here, this intervention appears in the model by decreasing the mosquito birth rate $\mu_o$, as shown in Eq 7:

$$\mu_o(C) = \mu_c(1 - \omega_o \, \theta(C - C_p)) \qquad (7)$$

where $\theta(\ldots)$ is the unitary step function, $\mu_c$ is the natural rate of birth/death of the mosquito, $C$ is the cumulative number of infected people, $C_p$ represents the cumulative amount of infected people needed to trigger the action and, $\omega_o$ is the parameter related to how efficient are the population and government actions in preventing the vector from laying the eggs, which led to decrease the mosquito birth rate $\mu_o$.

Fig 5 shows the distribution of the number of cases and the cumulative number of cases as a function time for different intensities of the mosquito birth rate reduction. Similar behavior as the fogging intervention is observed here. In Fig 5A, the number of cases reaches the peak of infected people sooner and then starts a decay in the number of cases per week. The total number of case drops to 85.3% when $\omega_o = 0.25$ presented in dashed red line and drops to 70.7% for $\omega_o = 0.5$ shown in the dotted yellow line when compared with the data without the

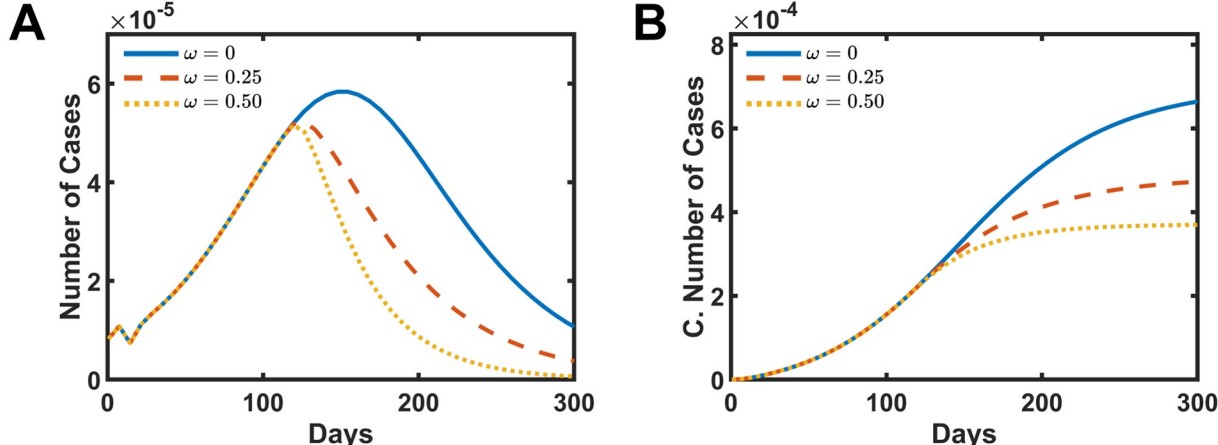

**Fig 4. Simulated intervention results in increasing the mosquitoes mortality rate by fogging action.** A and B present the number of cases and the cumulative number of infect people, respectively. The solid blue line is the simulation without intervention $\omega = 0.0$. The dashed red line presents the data for $\omega = 0.25$, and the dotted yellow shows the data for $\omega = 0.5$.

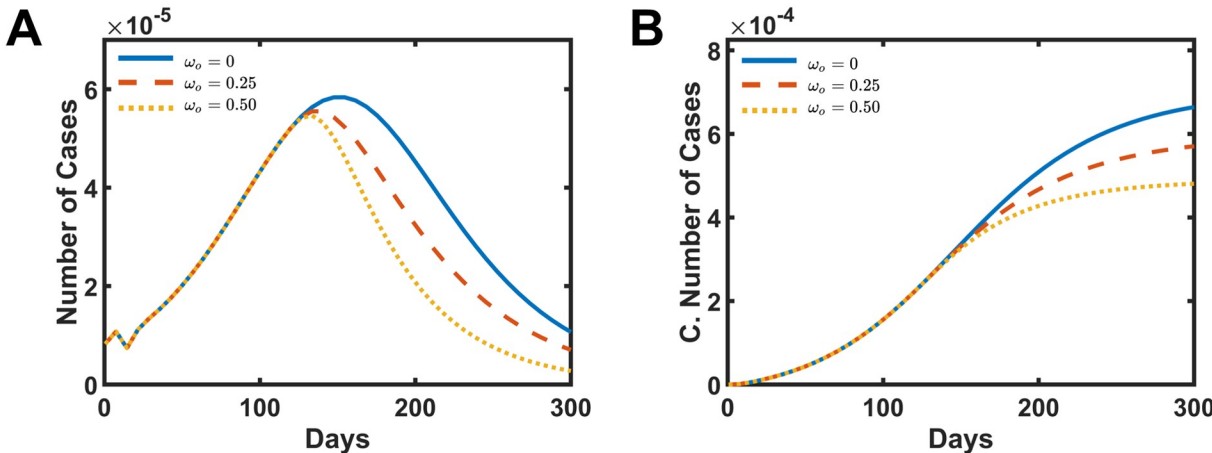

**Fig 5. Simulated intervention results in decreasing the vector birth rate by the action of removing the places where mosquitoes lay the eggs.** A and B present the number of cases and the cumulative number of infect people, respectively. The solid blue line is the simulation without intervention $\omega_o = 0.0$. The dashed red line presents the data for $\omega_o = 0.25$, and the dotted yellow shows the data for $\omega_o = 0.5$.

intervention $\omega_o = 0.0$ presented in Fig 5B. Although the behavior is similar to the fogging action, the response of decreasing the mosquito birth rate to the total number of cases is less efficient during the outbreak. It is worthwhile to mention that this kind of action, different from the fogging, can be inherited and passed to the following years and avoid new outbreaks to occurs in the future.

The third and the last studied intervention acts as the reduction of the rate in which infected humans transmit the disease to the mosquitoes. This effect can be associated as a quarantine action, isolating infected people or, more realistic, the usage of repellents by the infected human. Both strategies go in the direction of decreasing the contact between infected humans and the vector which is simulated using the Eq 8.

$$\beta_m(C) = \beta_c(1 - \epsilon \, \theta(C - C_p))\tag{8}$$

where $\theta(\ldots)$ is the unitary step function, $\beta_c$ is the natural rate at which humans infect mosquitoes, $C$ is the cumulative number of infected people, $C_p$ represents the cumulative amount of infected people needed to trigger the action and, $\epsilon$ is the parameter that modulates how intense is the decrease in the rate at which humans infect mosquitoes $\beta_m$.

Fig 6 presents the results for this last intervention. The curves in Fig 6A and 6B show a similar pattern, as observed in the other two previous actions. The number of cases shows a decay after the intervention starts for different intensities of $\epsilon$. In Fig 6B the total number of cases curves present results closer to the birth control intervention than the fogging action. For $\epsilon = 0.25$, the total number of cases decreased to 82.7% of the initial value, meanwhile, for $\epsilon = 0.5$ this number drops to 64.1%.

A combined simulation applying all the three interventions was carried out, and the results are presented in Fig 7. The combined intervention presents, as expected, the most effective strategy to decrease the number of infected people. The distribution curve of the number of cases per week shows more intense decay in Fig 7A. The total number of cases drops to 54.3% from the initial value when $\omega$, $\omega_o$, $\epsilon = 0.25$ and reduce to 43.6% for the parameters $\omega$, $\omega_o$, $\epsilon = 0.5$ in Fig 7B.

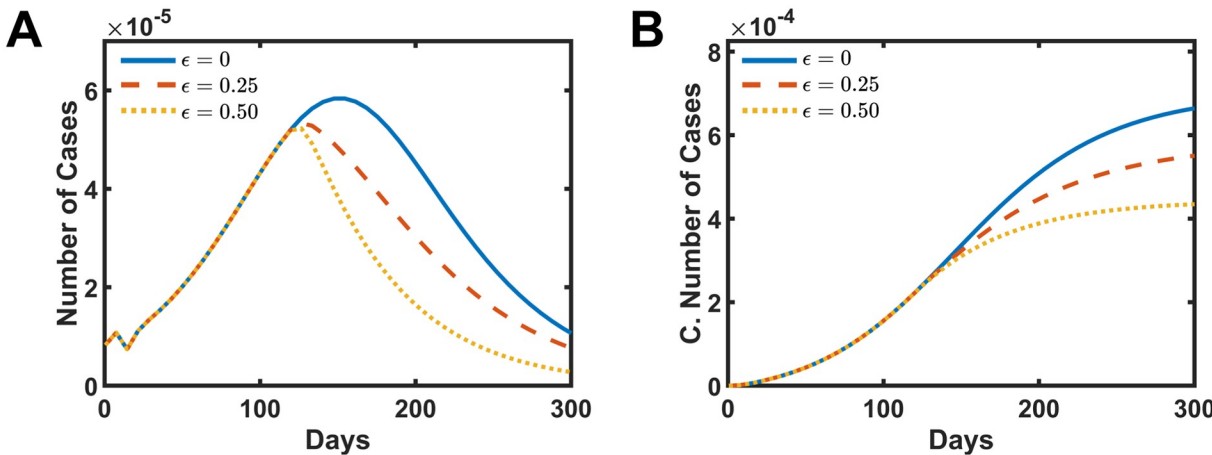

**Fig 6. Simulated intervention results in decreasing the rate in which humans transmit the disease to susceptible vectors.** A and B present the number of cases and the cumulative number of infect people, respectively. The solid blue line is the simulation without intervention $\epsilon = 0.0$. The dashed red line presents the data for $\epsilon = 0.25$, and the dotted yellow shows the data for $\epsilon = 0.5$.

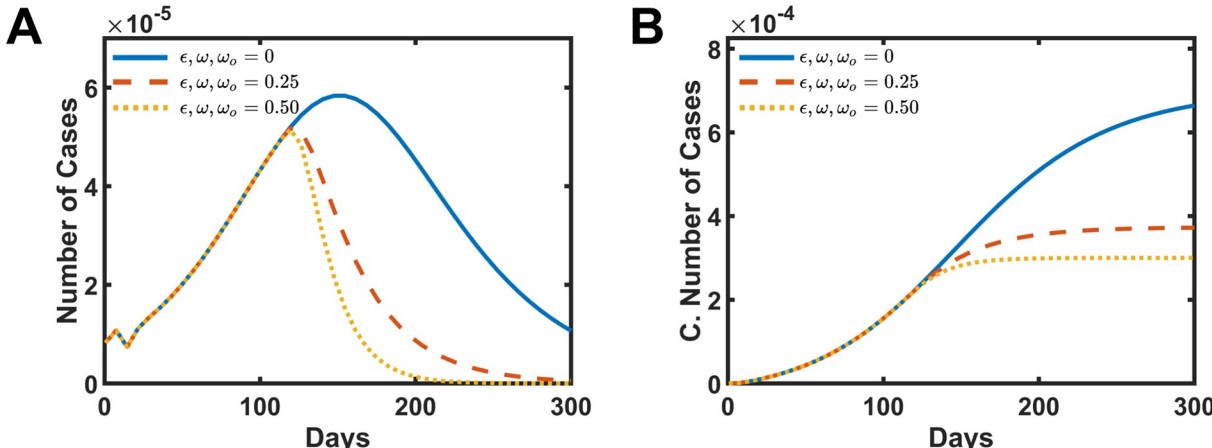

**Fig 7. Simulated intervention results by the combination of all the other three actions discussed in this work.** A and B present the number of cases and the cumulative number of infect people, respectively. The solid blue line is the simulation without intervention $\omega, \omega_o, \epsilon = 0.0$. The dashed red line presents the data for $\omega, \omega_o, \epsilon = 0.25$, and the dotted yellow shows the data for $\omega, \omega_o, \epsilon = 0.5$.

## Conclusion

The last three Chikungunya outbreaks in the city of Rio de Janeiro, Brazil, were modeled using the SEIR model and estimates the Basic Reproduction Number $R_o$ for the years 2016, 2018, and 2019. The simulations results register values greater than 1 for all of them, and 2019 is the most severe, even though the data was limited for the first six months. The calculation of $R_o$ gives a global overview of the impact and scale of the outbreak. Sensitivity analyses were performed to indicate, quantitatively, the importance of each parameter to the epidemic profile in different stages of the outbreak. A more detailed approach could take into account the number of infected people in each neighborhood with different sanitary conditions, and such details are not explored in this work. This study was expanded to include the Mayaro virus, which was reported as an emerging disease in South America [37, 39, 41]. Based on the assumption

that Mayaro and Chikungunya viruses have a similar spreading mechanism [37], since both viruses have the same vector [31, 36, 42, 43], we used parameters fitted from the Chikungunya outbreak from 2018 to estimate the $R_o^{MAYV}$ from Mayaro. The results indicate that Mayaro has the potential to be an epidemic disease in Rio de Janeiro with $R_o^{MAYV}$ values in a range of 1.18 and 3.51. Also, to possibly stop or at least decrease the intensity of an outbreak, three interventions strategies were proposed by modifying the basic equations of the SEIR model. These interventions are associated to the increase of the vector mortality rate by fogging techniques, the decrease of mosquito birth rate by decreasing the amount of places where the mosquito lay the eggs and, the decrease of the rate in which humans transmit the disease to mosquitoes by the isolation of infected people or the usage of repellent. Although those simulations do not retract real data, they can contribute to discussions about public and government policies directions.

## Supporting information

**S1 Data.**
(ZIP)

## Author Contributions

**Conceptualization:** Esteban Dodero-Rojas, Luiza G. Ferreira, Vitor B. P. Leite, José N. Onuchic, Vinícius G. Contessoto.

**Data curation:** Vinícius G. Contessoto.

**Formal analysis:** Esteban Dodero-Rojas, Vinícius G. Contessoto.

**Investigation:** Esteban Dodero-Rojas, Vinícius G. Contessoto.

**Methodology:** Esteban Dodero-Rojas, José N. Onuchic, Vinícius G. Contessoto.

**Project administration:** Vinícius G. Contessoto.

**Resources:** José N. Onuchic.

**Software:** Vinícius G. Contessoto.

**Supervision:** José N. Onuchic, Vinícius G. Contessoto.

**Validation:** Vinícius G. Contessoto.

**Visualization:** Vinícius G. Contessoto.

**Writing – original draft:** Esteban Dodero-Rojas, Luiza G. Ferreira, José N. Onuchic, Vinícius G. Contessoto.

**Writing – review & editing:** José N. Onuchic, Vinícius G. Contessoto.

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
