## [Decision Letter · Decision Letter 0]

7 Nov 2019

PONE-D-19-25070

Modeling Mayaro and Chikungunya Control Strategies in Rio de Janeiro Outbreaks

PLOS ONE

Dear Dr. Contessoto,

Thank you very much for submitting your manuscript "Modeling Mayaro and Chikungunya Control Strategies in Rio de Janeiro Outbreaks" (#PONE-D-19-25070) for review by PLOS ONE. As with all papers submitted to the journal, your manuscript was fully evaluated by academic editor (myself) and by independent peer reviewers. The reviewers appreciated the attention to an important health topic, but they raised substantial concerns about the paper that must be addressed before this manuscript can be accurately assessed for meeting the PLOS ONE criteria. Therefore, if you feel these issues can be adequately addressed, we invite you to submit a revised version of the manuscript that addresses the points raised during the review process. We can’t, of course, promise publication at that time.

We would appreciate receiving your revised manuscript by Dec 22 2019 11:59PM. To enhance the reproducibility of your results, we recommend that if applicable you deposit your laboratory protocols in protocols.io, where a protocol can be assigned its own identifier (DOI) such that it can be cited independently in the future. For instructions see: http://journals.plos.org/plosone/s/submission-guidelines#loc-laboratory-protocols

We look forward to receiving your revised manuscript.

Kind regards,

Abdallah M. Samy, PhD

Academic Editor

PLOS ONE

Journal Requirements:

1. Please note that PLOS ONE has specific guidelines on software sharing (http://journals.plos.org/plosone/s/materials-and-software-sharing#loc-sharing-software) for manuscripts whose main purpose is the description of a new software or software package. In this case, new software must conform to the Open Source Definition (https://opensource.org/docs/osd) and be deposited in an open software archive. Please see http://journals.plos.org/plosone/s/materials-and-software-sharing#loc-depositing-software for more information on depositing your software.

Reviewers' comments:

Reviewer's Responses to Questions

**Comments to the Author**

1. Is the manuscript technically sound, and do the data support the conclusions?

Reviewer #1: No

Reviewer #2: Yes

2. Has the statistical analysis been performed appropriately and rigorously? 

Reviewer #1: No

Reviewer #2: Yes

3. Have the authors made all data underlying the findings in their manuscript fully available?

Reviewer #1: Yes

Reviewer #2: Yes

4. Is the manuscript presented in an intelligible fashion and written in standard English?

Reviewer #1: Yes

Reviewer #2: Yes

5. Review Comments to the Author

Reviewer #1: Authors are presenting a mathematical epidemiology modeling approach to describe outbreaks of Chikungunya in Rio de Janerio, Brazil for the years 2016, 2018, and 2019. Parameters for the Chikungunya model are obtained from the published literature and unknown parameters are estimated using the best fit to the solved equation. Then, parameters for Mayaro virus are plugged in the equation in order to calculate the basic reproduction number for Mayaro in the same geographic region. Finally, for Chikungunya in the year 2018, the authors developed equations resembling three vector control strategies.

The article is well written and their modeling is a book approach to vector-borne disease mathematical modeling. Although well developed, the paper lacks a broader discussion of their findings or lack of them. First, the title of the manuscript is not reflecting what is done in the manuscript, specifically; intervention estimations are done only for Chikungunya in 2018, so Mayaro is never considered apart of the R0 estimation.

Considering the Mayaro model, authors are replacing Chikungunya parameters by those described for Mayaro and then presented results for this pathogen. Although this is valid, the manuscript lacks any other discussion on the topic, this is key in order to consider this manuscript as a scientific contribution worth to be published in a journal, otherwise, we can also fit the parameters for any vector-borne disease (let’s say, Eastern Equine encephalitis, West Nile, etc) and comment on the basic reproduction number. Mayaro specifically grants discussion regarding its life sylvatic/urban life cycle, its endemicity in Latin America (it has been present in the continent longer than Chikungunya), its reservoirs, among others. Further, from the infectious diseases dynamics perspective, is more similar to Yellow fever, thus, the taxonomic justification (Chikungunya and Mayaro are alphaviruses) is short to explain its dynamics in any particular region.

The model is also lacking uncertainty measures. Parameters for the Chikungunya model include ranges for the human and mosquito latent periods. Are the authors using the median of this range? Which are the uncertainties associated with their simulated best-fit distribution of cases presented in Fig 1? Which are the most sensitive parameters in their model? (Sensitivity analysis?)

I believe, the paper could be very useful in a journal considering ‘teaching corners’ since its model implementation is not novel enough but is well explained and developed. It can also be deployed with its code for further teaching applicabilities.

Reviewer #2: It is a well structured paper, clear and easy to read. Authors recognize that models provide insights in the behaviour of epidemics and help further discussions of goverment policies, but do not show real data.

6. PLOS authors have the option to publish the peer review history of their article (what does this mean?). If published, this will include your full peer review and any attached files.

Reviewer #1: No

Reviewer #2: Yes: Carolina Ramírez-Santana

---

## [Author Response · Author response to Decision Letter 0]

17 Dec 2019

The response and comments letter are attached with the manuscript files.

---

## [Editor Report · Decision Letter 1]

3 Jan 2020

Modeling Chikungunya Control Strategies and Mayaro Potential Outbreak in the City of Rio de Janeiro

PONE-D-19-25070R1

Dear Dr. Contessoto,

We are pleased to inform you that your manuscript has been judged scientifically suitable for publication and will be formally accepted for publication once it complies with all outstanding technical requirements.

With kind regards,

Abdallah M. Samy, PhD

Academic Editor

PLOS ONE

---

## [Editor Report · Acceptance letter]

7 Jan 2020

PONE-D-19-25070R1 

Modeling Chikungunya Control Strategies and Mayaro Potential Outbreak in the City of Rio de Janeiro 

Dear Dr. Contessoto:

I am pleased to inform you that your manuscript has been deemed suitable for publication in PLOS ONE. Congratulations! Your manuscript is now with our production department. 

With kind regards,

on behalf of

Dr. Abdallah M. Samy 

Academic Editor

PLOS ONE